# Oral Reflection Tasks: Advanced Spanish L2 Learner Insights on Emergency Remote Teaching Assessment Practices in a Higher Education Context

**Ana Maria Ducasse** 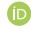

**Abstract:** This paper reports on a small-scale study that is the first to explore Advanced Spanish L2 learners' personal awareness of their language and culture learning through e-assessment tasks in an Emergency Remote Teaching (ERT) context, mediated by five task-specific, individual spoken reflections. The value of reflection in education, particularly for L2 writing and distance learning, has been explored in different modalities, e.g., individual spoken reflection and group spoken reflection. Building on previous research, this study explores a group of advanced Spanish L2 learners (n = 25) reflecting on five multi-modal e-assessments through individually assessed oral audio-recorded post-assessment reflection tasks (n = 125). A thematic content analysis applied to transcriptions yields findings from a pedagogical perspective on language learning, completing assessments and personal affective responses. The learners' candid and explicit orientations towards various types of multimodal language-learning e-assessment tasks offer instructors information on learners' awareness of classroom-based assessment tasks being enablers for individual learning goals.

**Keywords:** e-assessment; assessment in COVID-19; synchronous mode; Spanish teaching; Emergency Remote Teaching (ERT); L2 learner reflection; learner assessment literacy

## 1. Introduction

Language testing and assessment contexts are characterized by assessment practices compatible with the social and pedagogical values and beliefs of the local learning community. One such context comprised the implementation of unplanned "Emergency Remote Teaching" (ERT) in response to the COVID-19 pandemic (Hodges et al. 2020) which subsequently affected all stakeholders from local to global contexts in the migration to e-learning.

Technical issues were to affect the assessment practices of most stakeholders who were using diverse platforms for the first time. For each course delivered, practitioners rationalized content and adapted assessments and marking criteria to meet revised learning outcomes. This extraordinary context affords an opportunity to explore the practices, perceptions, and knowledge about learning and assessment (Scarino 2013) that emanated from the changed conditions due to the pandemic. By stealth, COVID-19 provoked a general overhaul of teaching away from print media to digital technologies. Classrooms had been reticent to reflect widespread adoption of digital ways of working (Henning 2020). Although the focus on multimedia-based student-centered learning with new e-assessments had been looming, it had met resistance and was long overdue.

In language teaching, multi-modal digital classroom assessments, or MDCAs, refer to any assessment practices designed by a teacher that require students to combine two or more representational modes using digital technology (Henning 2020). In other words, a digital version of integrated assessment tasks, i.e., video, and online reading with digital photography input, to digital writing output; online reading with digital photography and video input, to digital oral presentation output or online reading with digital photography and audio listening, to oral digitally recorded output. These changes aimed at more

generic skills, such as 21st century IT skills, group work, global competence, through topics explored also included the fortuitous reflection task, which is pivotal to this enquiry.

Language student orientation to MDCAs bears exploring and one way of gleaning new knowledge is through individual reflection which is widely used in educational research for different purposes, one of these being assessment. There are various ways to promote reflection on assessment, e.g., reflective journals, reflective pre-assessments, and reflective post-assessments (Tanner 2012) and it has been suggested that the amount and type of reflection is influenced by the discourse mode, the task, the participants, and power dynamics (Farr and Riordan 2012).

A reflection can serve as a conduit for participants to engage in deeper level thinking that might involve, inter alia, recalling, relating, reasoning, hypothesis generating or testing, and solution generation after completion of a task (Betts 2004; Brockbank et al. 2002). In the context of higher education language learning, the reason for asking test takers to reflect on assessment situations arises from research suggesting that test-takers reflecting on a previous task might affect their performance in subsequent tasks (e.g., Huang 2009; Swain et al. 2009). Important questions remain in relation to the impact of reflection on learning and thus its educational value, which needs further consideration (Farr and Riordan 2012). In addition, there remains a lack of contextualized examples or practical guidance for practitioners to conceptualize reflection and support the development of reflective capacity in students (Mann et al. 2009).

Of particular interest are learner perceptions of the social and pedagogical changes that ensued from teacher modifications to assessment practices in response to ERT. The study reported on below provides an exploration of learners' orientation towards their changed and evolving local learning community, through guided, recorded and marked oral reflection tasks, based on assessments learners submitted during the ERT implementation period. The analysis of these student reflections adds to the body of knowledge on pedagogically sound classroom-based assessment methods in this context, e.g., Chen and Sato (2021) and will interest language teachers, online language course developers and the instructional designers who work with practitioners to enhance online pedagogy. The students' personal reflections on their assessment tasks correspond to Brookfield's lens of reflection on one's own experience (Brookfield 2017).

*Rationale for Study*

In this exploration of a university-level advanced Spanish class, spoken data elicited from guided learner reflections offer an opportunity to examine evaluation foci innovatively in order to interrogate language students' response to e-assessments. The remote assessment conditions that threatened test security meant that instead of incremental introduction of multimodal e-assessments, these were introduced at once to offer authentic text assessments by utilising e-modalities to replace pen and paper. There is as of yet no research investigating how students oriented towards advanced L2 language e-assessments during the ERT brought on by the pandemic crisis. Such research provides evidence that students value the qualities they attribute towards e-assessments, as they compare their experiences to an e-assessment-free context known to them from previous education in F2F non-remote courses.

The main aim of the study was to uncover student orientation to e-assessment through reflective learning during an oral e-assessment task in order to understand how the innovations to assessment were received by the students new to working in a remote context which would be beneficial for course designers and language instructors. Language students with no previous experience of ERT language learning, offer perspectives on experiences of e-assessment and what they considered conducive to learning which builds on a European study of English learners and their experiences of assessment (Vogt et al. 2020).

Reflection is a transferrable competence in lifelong learning where learners feel in control over future actions. Reflective learning is also a 21st century skill that grows awareness to help adapt to the challenges of technology and ever-changing environments

(Colomer et al. 2020). Reflective learning from an oral reflection e-assessment task means "contemplating general or specific contexts, practices, scenarios, problems or issues directly or peripherally relevant to the discipline of study" (Farr and Riordan 2012, p. 129). Since students have not completed assessments in these modalities previously, a candid account is expected because they were 'surprised' when ERT was implemented two weeks into their semester. Although the original course assessments listed in the university online course outline would have stated similar intended learning outcomes (ILOs), these were intended for traditional pen and paper, F2F, in-class assessments. The outcome from the analysis of the oral reflection tasks on these e-assessments is expected to render a rich and nuanced picture from which to gain insight into how students approached the 'adapted' assessments tasks, what they perceived to have learned for future assessments from completing the tasks and their affective responses towards the course assessments.

To this end, the following questions will be explored:

- RQ 1 What did the students report learning about from the language and culture content through completing assessments?
- RQ 2 What qualities of the e-assessments did the students value as they executed the tasks?
- RQ 3 How did students orient towards the e-assessment tasks in the reflection?

## 2. Materials and Methods

### 2.1. Research Context

RMIT University is a public institution in Australia where languages are traditionally taught face-to-face (F2F) so in the semesters when the pandemic affected the Spanish program, social distancing meant lecturers and learners were forced into ERT regardless of their preparedness. Hence, ERT is a crisis by-product to temporarily shift a F2F course to the fully online mode without rendering the teacher or course designer sufficient planning time to develop a robust online course (Whittle et al. 2020).

The fast transition during the move to ERT meant that across the board in varied educational contexts the usual planned development process for effective online course delivery did not include conversations around enhanced pedagogy with online developers and instructional designers (Gacs et al. 2020). In sum, although the courses appeared 'online', they were unusual, and it would be unfair to appraise them as one would established online courses.

The university-wide courses at RMIT, before the implementation of ERT, were supported by the CANVAS Learning Management System which offered Collaborate as platform for synchronous web-enabled F2F teaching and supported class-paced learning. The teacher/student ratio was 25/1 with a high participation rate for the Spanish instruction that commenced with two two-hour classes F2F. From week 3 of the twelve-week semester an Emergency Remote Teaching response was implemented. The students' role was to listen and/or read texts in preparation for class, then collaborate with peers synchronously on tasks in breakout groups using the prepared materials. Finally, they would interact orally or in written form via a chat or class whiteboard with the lecturer and the class group. The types of pedagogy comprised practices of all types required for language acquisition: explanatory, supported by slides focused on course content for language learning outcomes; real-time collaborative tasks; and expository, e.g., students delivering pre-recorded group oral presentation videos to other students on cultural topics for peer review.

Spanish courses at RMIT are designed following the CEFR with an acknowledgement of the companion volume 2020, which includes updated ILOs to include language skills particularly for online and digital contexts. From beginner to advanced, with entry points determined by a placement test, students are predominantly enrolled in International Studies or International Business. Four semesters of language study are the minimum graduation requirement for International Studies and Spanish is one of four language sequences taught from beginner level through to advanced. All class materials are archived in CANVAS for viewing during the entirety of the course and include elements of au-

thentic dialogues and texts, problem-solving, task-supported activities, and lesson-related explanations, vocabulary lists and grammar summaries. After each semester, materials are updated to reflect students' interests and relevant recent developments in the Spanish speaking world.

The teacher/researcher of the present study adapted or redesigned both the summative and the formative assessments for ERT delivery within the Learning Management System. Consideration was given to authentic performance-driven assessment possibilities to demonstrate learning outcomes working within the "remote" delivery mode. Bearing in mind that evaluating language proficiency online is a messy task (Blake 2015) the assessment criteria were tweaked for the new context. (The adaptation process of the tasks and the criteria falls beyond the scope of this paper). This response aimed to guard against the inevitable threat of test security issues, which would benefit cheats and discriminate against honest students (Koris and Pál 2021). Despite the crisis, the redesigned assessments and marking grids utilized more authentic online materials compatible with increased remote security delivery and geared towards MDCA tasks. Unsurprisingly, however, the marking load increased in line with the quantity of spoken and written proficiency assessments for formative feedback before grading.

Students took and submitted assessments synchronously. After the initial formative course assessment task (quiz 1, see Table 1) four weeks into semester to identify students at risk (despite a compulsory diagnostic level test), the role of the remaining assessments was to provide students and teachers with information about learning and to contribute to grades via formative and summative assessments. Feedback was predominantly teacher focused for proficiency assessments; some drafting was used for writing tasks and automated feedback was also used for the final culture topic reading and listening comprehension.

**Table 1.** Task distribution over 12-week semester.

| Formative | Week | Summative |
|---|---|---|
| Quiz 1 Reading/written summary | 4 | |
| 5 × Oral reflection tasks | 4–14 | |
| Quiz 2 Listening/Speaking | 6 | |
| | 8 | Video/reading/written report |
| Peer review/feedback | 10 | Group oral video PPT presentation |
| | 12 | Quiz Final Reading/listening/writing |
| | 14 | Reading/writing Essays (UG+PG) |

Assessment practices are explored here from a single teacher/researcher and a student perspective. Online learning and assessment experiences in a broad education context can evaluated in terms of "success" as defined by stakeholders. From a specifically learner perspective in this study, elements of course success might include, but are not limited, to motivation for language and culture learning, engagement with the e-course material and e-assessment tasks and personal interest in the topics. These elements connected to learner success will serve as insightful foci for exploration and discussion.

### 2.2. Course Assessment Tasks

New MDCA tasks and rubrics were developed or adapted by the teacher/researcher after the course commencement in reaction to the sudden change from F2F classroom context to synchronous class teaching in the ERT environment. Tasks were considered formative or summative and were set fortnightly throughout the semester (see Table 1). Tasks included digitally recorded oral presentations, integrated tasks (i.e., video plus reading input with written summary (quiz 1) or audio and reading input with spoken response (quiz 2).

The assessed reflection tasks (Appendix A) and marking rubric (Appendix B) offered students an opportunity to describe their perceived learning outcomes on five MDCA assessments completed during the semester for 10% of total course marks. When making

personal reflections, it is important for students to feel secure and in a non-threatening environment as they express sensitive insights on their self-perception or perception of their peers (Stocks and Trevitt 2016). Bearing this in mind, students were to submit five audio reflections each (and as per the rubric, select their 'best' one for marking).

The rubric was designed by the teacher/researcher as a tool for the students to production the reflection and for the assessor to award marks. Just as rubrics guide students on assessor expectations in other assessments, Cheng and Chan (2019) experimented by incorporating a rubric as a resource for reflection. They found the rubric promoted students' level of reflection. The rubric was written in simple clear language based on issues reported on student use and understanding of reflection language as reported in the Alsina et al.'s (2019) Narrative Assessment Reflection Rubric validation study in higher education.

### 2.3. Participants

The B2 level comprises students for the most part in their third year of language study frequently following travel, in-country study or international placements abroad. Heritage speakers who studied at high school and returning travellers with highly developed oracy and literacy skills in the language join them in their first year, along with international exchange students with previous Spanish studies. A small number of post-graduate students take languages relevant to their studies, e.g., International Development or International Relations. The total number of students enrolled across two B2 classes during the COVID-19 lockdown in Melbourne was 29 students and the study participants (n = 25) were those who submitted the oral reflection assessment tasks which included five post-graduate students.

In the ERT context, the participants reflect the students who were tenacious and persevered in the ERT environment with self-motivation and drive, some returning home from exchange and maintaining studies from home and other international students following classes without leaving home due to international border closures. The study has a limitation in that it does not include the views of those who at the time of ERT did not have the technical facilities, i.e., Wi-Fi, laptop, or suitable study environment during lockdown or the economic means for stability and livelihood as reported by, for example, Drane et al. (2020), Mupenzi et al. (2020) or O'Shea et al. (2021). At the time of writing, RMIT Melbourne is in its third semester of strict lockdown over 18 months, the highest number of days and weeks in the world.

### 2.4. Data

The data comprise student oral reflections recorded after multimodal e-assessment tasks. The teacher researcher accessed the oral reflections with ethics clearance and any student's reflections quoted in this paper consented to the anonymous use of their transcribed and translated reflections for research purposes.

Participants selected five completed course assessments, then submitted an oral reflection for each as a one-minute audio in Spanish. The data comprised the transcription of up to five oral reflections per student.

### 2.5. Data Collection

With ethics approval that allowed assessment tasks to be analysed for research, students were contacted via email. They could opt in for their reflections to be analysed anonymously for research, with no further participation required of them. Data collection took place after marks were submitted and accepted by the assessment committee, although the oral reflections had taken place during the semester, after assessment tasks, starting in week four.

A total of 116 transcriptions from 25 participants were usable from a possible 125, i.e., some repeated a submission or submitted only 4 tasks, other students spoke completely off topic on submissions which were not analysed for not achieving successful task completion due to irrelevance. Partly irrelevant submissions were included in the sample, i.e., if following the rubric, students left out parts or were completely off topic and not following

the guide for a section of the reflection, the remainder could still be analysed and from the frequency in Table 2, which shows the number of reflections per assessment, we can see that the formative, first two multimodal assessments, the Quiz 1 reading/writing summary and Quiz 2 listening/speaking, make up a third, or just over 30%, of the reflection submissions. The summative culture-based research essay and the group oral presentations were commented on by most students, thus making up around 40% of the reflection submissions. This leaves the summative language and culture final and the integrated video listening task also on a cultural topic making up the shortfall. Two students chose to reflect on the reflection tasks that were not a listed option.

**Table 2.** Transcribed number of student oral reflections and frequency in data.

| Multimodal Task Type | N Reflections | Frequency |
|---|---|---|
| Quiz 1 Reading/writing summary | 18 | 18/15.51% |
| Quiz 2 Listening/Speaking | 19 | 19/16.38% |
| Online reading and researching/Essay | 23 | 23/19.83% |
| Group oral presentation | 23 | 23/19.83% |
| Quiz Final Reading/listening/writing | 17 | 17/14.66% |
| Video/reading/writing report | 14 | 14/12.07% |
| Oral reflection tasks | 2 | 2/1.72% |

*2.6. Analysis of the Transcribed Oral Reflection Tasks*

The completed reflection audios files were downloaded from the learning management system, transcribed and then analysed for thematic content. The reflection tasks were automatically transcribed, then checked and cleaned of hesitations, repetitions and repairs prior to segmentation and coding. The first step in the analysis was to develop an approach to segmentation by scanning the data. As it was content and thematic analysis, meaning-based units of analysis were used. The researcher began segmenting and colour coding the ideas expressed for 20% of the transcriptions (5 in total), selecting those that appeared longer, in that more words were spoken per minute by the students, indicating perhaps more fluency or higher proficiency. The segmented data were entered into columns of an Excel spreadsheet, with identifying data such as student IDs and the task in additional columns.

The next step involved an attempt to specify a set of coding categories using grounded theory and open coding, using an inductive approach to conceptualise and classify the patterns in the data as applied, e.g., in Ducasse and Brown (2009) and Brown and Ducasse (2019). Defining categories requires repeated data reduction, rearranging and recoding until the balance is found between too many and too broad categories (Green 1998).

After re-reading the entire dataset, tentative categories based on some distinctions emerged that could clearly be seen in the data for several reasons. The proficient students had followed the guide (Appendix A), so they spoke about the tasks, their learning and how they felt about them. This divided the reflective comments easily across these broad categories with other subcategories subsumed.

One general difficulty was that some reflection comments were very broad and, at times, unspecific, showing that students had not followed the task guide (Appendix A). If nothing was said about any of the three overarching categories, those reflections were discounted for off-task irrelevance and were not included. The example below of an off-task irrelevance shows the student reflects only generally on the learning experience and the change to ERT but does not mention the summary task completed prior to submitting the reflection. They mention that things improved for them as time went on and in fact for the remainder of the reflections, the student followed the guide and rubric, so the work was included in the data as a 'reflection'.

Translated from a transcription of spoken Spanish that has not been included as data for being off task: "*The change to online classes was very difficult, in the one hand I didn't know how to use collaborate and when I did it was very hard to find a good internet connection. On the other hand, the transition to working online at RMIT at the same time had been a challenge and*

*frequently I found myself too exhausted ager work when it was time to start class. I also felt that as if the other students were more advanced than I. It was impossible to see or read any facial expression and this did not help much either; even though this began to improve, as time went by everyone slowly kept adapting, but it wasn't as good as it could have been."*

As different aspects of the reflection such as appraisals or affective responses were sometimes syntactically interwoven with other categories and hence not separable into distinct segments, codes were assigned for as many themes as were identified.

The codes were entered into cells alongside the extracts in the Excel spreadsheet. The coding process was iterative, sorting sub-themes into three larger categories (see Table 3) for reflective comments on doing the task, learning the language and culture topics and reflecting on themselves, their peers and their feelings via appraisals or affective responses. These categories seemed clear once the off-task submissions were discounted. Some units did not refer to aspects of specific assessments but about the course generally. Those comments were coded as 'other' and subsequently deleted from the analysis.

**Table 3.** Coding themes, subthemes, and definitions.

| Theme | Sub-Theme | Definition |
|---|---|---|
| | The content | -topics from Spanish speaking world |
| 'Learning' | The language | -vocabulary/grammar/skills/Accents and speed |
| 'Doing assessment' | The task | -preparation task/plan for next time/connection to other tasks<br>-problem solving/tackling challenges<br>-e-learning/researching/technical skills |
| 'Reflecting on learning and doing assessment' | The affective and personal<br>The interpersonal | -awareness of self/sense of improvement/personal fulfilment<br>-affective response, e.g., enjoyment distress frustration difficulty<br>-appraisal, e.g., like/dislike, good/bad, usefulness/relevance<br>-collaboration/learning with peers or from peers<br>-future plans: studies, work, family |

## 3. Results

Three overarching independent categories could be easily and reliably identified, each concerned with a discrete focus of the reflection and together accounted for almost all the data. These comprised 'Learning', 'Doing assessment' and 'Reflecting on learning and doing assessment'. Table 3 lists the reflection focus categories with an expansion into sub-themes that arose from further analysis within the main categories. The 'learning' theme divides into comments on the content and the language. The 'doing assessment' theme is broadly defined by all things related to the task and has no subtheme. The reflecting on learning and doing assessment theme divided into two overarching themes, i.e., the personal and affective and the interpersonal. The personal and affective sub-theme subsumed many examples to define it, including self-awareness, affective responses and appraisal.

The distribution of number and frequency of comments by task and theme across the tasks in Table 4 shows the group oral presentation attracted the most comments and the integrated reading video and report writing attracted the least. However, these frequencies should not be interpreted as an indication of their relative importance or salience, given that in the reflection tasks, students select from many thoughts and summarize ideas to be succinct in their one-minute reflection. Additionally, as they record the reflection in real time, they may be more comfortable with articulating some aspects of their reflection depending on their fluency and proficiency. The categories tend, unsurprisingly, to match elements from the task guide and rating rubric (Appendices A and B) because the reflection task was assessed.

**Table 4.** Distribution of number and frequency of comments by task, total and theme.

| Assessment Task | Total Comment by Task N/% | Theme 1 Lang./Culture 'Learning' Comment N/% | Theme 2 Task 'Doing . . . ' Comment N/% | Theme 3 Personal 'Reflecting . . . ' Comment N/% |
|---|---|---|---|---|
| Quiz 1 Reading/writing summary | 46/16.67% | 16/35.55% | 14/31.11% | 16/35.55% |
| Quiz 2 Listening/Speaking | 47/17.02% | 14/29.79% | 15/31.91% | 18/38.29% |
| Video/Reading/writing report | 33/11.96% | 11/33.33% | 11/33.33% | 11/33.33% |
| Group oral presentation | 58/21.02% | 15/25.86% | 21/36.20% | 22/37.93% |
| Reading/writing Essays | 47/17.03% | 14/29.79% | 16/34.04% | 17/36.17% |
| Oral reflection tasks | 06/02.17% | 02/33.33% | 02/33.33% | 02/33.33% |
| Quiz Final Reading/listening/grammar/vocab | 39/14.13% | 11/28.20% | 15/38.46% | 13/33.33% |
| Totals | 276 | 83 | 94 | 99 |

During the analysis, before the calculations, a hypothesis was forming that the oral reflection tasks were encouraging students to express feelings, as they were coming across strongly as an opening exclamation in a way that would not be found in a written reflection, e.g., the Hill and Ducasse (2020) study, where level C1 students wrote 10 × 300-word formal and academic reflections in Spanish L2 on their learning after the construction of a reflection rubric in class with their lecturer. It appeared to be closely followed by comments and observations on newness of the e-assessment tasks and, although the learning had to be commented on as part of the task fulfilment, it did not receive such a prominent place. It transpired that the frequency of the totals coincided with that hunch, although with very close margins. The personal 'reflecting on learning and assessment' category attracted a 35.87% rate of comment, the 'doing assessment' category landed a 34.06% frequency, and finally, the language and culture content 'learning' category, which was not that far behind, received the remainder of the comments at a 30.07% frequency.

In the remainder of this section, each of the categories is detailed by research questions, i.e., on learning, doing, reflecting with, where relevant, a description of the more detailed orientations found within them along with illustrative examples, with translation and the task type (from Tables 2 and 4) in brackets.

Examples are direct quotes from students who signed up through ethics email recruitment for their assessments to be used for research anonymously (n = 9). They form part of those whose data are included in the study as participants (n = 25), but whose comments are reduced to frequencies in the reporting. As quotes from the oral reflection task are in Spanish, an English translation by the researcher follows with added context where required.

- RQ 1 What did the students report learning about from the language and culture content through completing assessments?

### 3.1. Theme 1: Learning Language and Culture through E-Assessment Tasks

Comments in this category addressed the topical and language content of the assessments. While some comments were very general about the e-assessments, others alluded to more specific aspects of the intended learning outcomes of the course variously encompassing the following: the personal relevance and interest in the content (Extract 1), students referring to vocabulary learning (Extracts 2 and 3), and new structures (Extracts 1 and 4). Other comments focused on listening tasks referred to speed (Extract 6) and accent (Extracts 5 and 6).

3.1.1. Subtheme: The Language

*Extract 1* (final quiz, H): Durante el curso aprendí muchas estructuras gramaticales y y estoy emocionada para usar mis aprendizajes en este, pero estuvo las PERIFASIS VERBALES en la segunda semana que se quedó conmigo porque fue muy temprano en

el semestre y porque lleve dos años sin estudiar español. Estuve luchando con algunas estructuras. Ese tema a mí introdujo a varias estructuras que podía usar en mis ensayos o cuando hablo. En mis viajes, negocios, y negocios de empresas [translated name of a course] también.

*During the Course I learned many grammatical structures and I am excited to use my learning of these, but it was the phrasal verbs in the second week that stuck with me because it was early in the semester at it had been two years since I'd studied Spanish. I was struggling with some structures. This topic [final quiz] introduced me to various structure that I can use in my essays and when I talk when travelling, for business and for business studies as well.*

*Extract 2* (V&R/Report, E): Después de esto, pude sentirme más seguro en español porque había explorado una nueva gama de palabras y temas que no conocía antes.

*After this I felt more secure in Spanish because I had explored a new range of words and topics that I didn't know before.*

*Extract 3* (Summary, E) Una cosa principal que aprendí en esa tarea fue cómo encontrar nuevas formas de decir lo mismo para ampliar mi vocabulario.

*The main thing I learned in this task was how to find new ways of saying the same thing to broaden my vocabulary.*

*Extract 4* (List/Speak, E): Y yo sé que hice muchos errores con conjugaciones de verbos en mis respuestas, aunque no era imposible.

*I made a lot more errors conjugating verbs in my answers, but it wasn't impossible [the task].*

*Extract 5* (quiz 2/J) Desde mi punto de vista, los exámenes de este curso de español fueron muy importante para mi aprendizaje. Antes de empezar este semestre, yo conocía nada más que el acento argentino, precisamente el acento porteño, porque mi mamá es de Buenos Aires. Gracias a los exámenes, particularmente el examen número 2 o el examen de audio, me pude familiarizar con otros acentos como el acento español, por ejemplo, sabiendo que en España hay varios acentos bien diferenciados.

*From my point of view the assessments in this course were important for my learning. Before starting this semester, I only knew the Argentinian accent, actually* Porteño, *because my mother is from Buenos Aires. thanks to the exams particularly exam 2 I familiarized myself with Spanish accents knowing that in Spanish there are various well marked accents.*

*Extract 6* (quiz 2/S) Mis habilidades auditivas son buenas en mi opinión, pero cuando las personas empiezan a hablar muy rápido se convierte en una actividad muy duro para mí. Creo que la cosa que me da las más dificultades fueron los 'tonos' de las personas que hablaron. Me encuentro dificultades con 'tonos' de Suramérica. Especialmente los argentinos y uruguayos.

*My listening skills are good in my opinion, but when people start to speak fast it becomes a hard task for me. I think what gives me the most trouble is the accents of the people speaking. I have difficulties with South American accents specially the Argentinian and Uruguayan.*

### 3.1.2. Subtheme: The Content

Comments in this category contained references to learning through the cultural content in the course (Extracts 7 and 8).

*Extract 7* (Cult./Pres., A): Cuando yo descubrí que mi presentación fue hacer sobre Guatemala, no fui feliz por. Principalmente porque tengo ideas y un mejor entendimiento sobre otros países y no supe unas cosas sobre Guatemala. Esto cambió mi mente, porque yo supe que este país pequeño sea rico y tenga muchas cosas fascinantes.

*When I discovered my presentation would be on Guatemala, I wasn't happy mainly because I had ideas and better understanding of other countries and I didn't know a thing about Guatemala. It changed my mind because I found out that this small country was rich in fascinating things.*

*Extract 8* (Cult/Essay, S): El ensayo cultural fue la culminación de mi trabajo del semestre. Me sentí feliz al ver mi cambio en mi capacidad de expresarme. Antes yo dependía de recursos en inglés para buscar información nueva y este proceso tomaba más tiempo y me hacía usar las expresiones en inglés. También estaba restringido por mi uso de traductores de oraciones básicas para formar mis pensamientos. En mi investigación

de este ensayo yo pude leer textos en español con confianza. aquel método significó que aprendí expresiones naturales de español y no necesité traducirlas. Mi forma de pensar cambio ya que yo dejé mi dependencia en inglés y en los traductores. Yo gané la confianza para obtener contenidos en español y expresar mis ideas en español directamente sin el filtro de inglés. Podré usar este aprendizaje para acceder a más contenido en español y hablar con más fluidez.

*The culture essay was the culmination of my work this semester. I felt happy to see the change in my capacity of expression. Before I depended on English resources to find new information and this process took longer and made me use English expressions. I was also restricted using basic phrases through translation to shape my thoughts. In the research for this essay, I could read texts in Spanish with confidence. That method meant I learned natural Spanish expression and I didn't need to translate them. My way of thinking changed by dropping my dependence on English and translation tools. I gained confidence in locating content in Spanish and expressing my ideas directly in Spanish without the filter of English. I can use this learning to access more content in Spanish and speak more fluently.*

- RQ 2 What qualities of the e-assessments did the students value as they executed the tasks?

*3.2. Theme 2: Student Reflections on 'Doing the Assessments'*

Comments in this category contained references to challenges for completing the e-assessment tasks (Extracts 9 and 10) and include problem solving as a sub-themes.

*Extract 9* (List./Speak, S): Necesite repetir las pistas de audio muchas veces para comprender completamente que dijeron, pero al final terminé el ejercicio sin problema.

*I needed to repeat the tracks many times to understand completely what they said but I finished the task without problems.*

*Extract 10* (List./Speak, C): Aprendí que cuando escuchaba español hablado en las pruebas, necesito cerrar mis ojos y concentrarme sin quedarme atascado en las palabras individuales. Esto me ayudó con la escucha en la prueba final y ahora intentaré ver más películas en español y música para mejorar mi capacidad auditiva.

*I learned that when I listened to spoken Spanish in the tests, I need to close my eyes and concentrate without getting stuck on the individual words. that helped me with the final listening and now I will try to see films and music to improve my listening proficiency.*

3.2.1. Subtheme: E-Learning/Researching/Technical Skills

Comments in this subtheme correspond to strategies for tackling challenges (Extracts 14 and 16), technical skills in IT (Extracts 12 and 17) and e-researching (Extracts 11, 13 and 15).

*Extract 11* (Summary, E): me gusto esa tarea porque me obligo a explorar noticias españolas y fuentes en internet que no hago suficiente. Y desde entonces me he empezado a leer y ver más con los medios españoles.

*I like this task because it forced me to explore Spanish news and internet sources that I don't do enough of. And since them I have begun to read more Spanish media.*

*Extract 12* (Cult./Pres., D): Aprendí diferentes formas de hacer presentaciones de forma creativa en las que el público está estaría más interesado en lo que estamos diciendo.

*I learned different ways of doing presentation in a creative way so the audience ismore interested in what we are saying.*

*Extract 13* (Cult./Essay, J): Cumpliendo con mi deber, aprendí a hacer mis búsquedas directamente en castellano.

*To complete the task, I learned to do searches directly in Spanish.*

*Extract 14* (Cult./Essay, S): Fue la primera vez que hice investigaciones con fuentes en español durante que leí las fuentes como noticias, revistas etcétera. Me di cuenta de que puedo comprender la mayoría de los artículos y palabras.

*It was the first time that I researched with Spanish sources while reading news and journalss etc. I realised that I could understand most articles and words.*

*Extract 15* (Cult./Essay, C): Me resultó difícil entender los textos académicos en español, pero me orientaron sobre cómo escribir de esa manera por mi propia obra.

*I found it hard to understand Spanish Academic texts, but they helped me with writing my own work.*

*Extract 16* (Cult./Essay, J): Gracias al ensayo sobre la gastronomía y el medio ambiente aprendí y logre escribir y sintetizar en castellano. Elegí una mata del altiplano andino, la coca o la coca boliviana.

*Thanks to the essay on gastronomy and the environment I learned to write and synthesise in Spanish. I chose a plant from the Bolivian plains, coca or Bolivian coca.*

*Extract 17* (Cult./Pres, S: Yo pasé tanto tiempo para investigar y hacer una presentación buena pero por los problemas de tecnología yo necesitaba hacer la presentación de nuevo en muchas formas y plataformas. Estuve muy triste que fuera de mi control el video no funcionó en el día. No creo que esto cambia mis experiencias de español porque tuve que pensar tanto en los horrores de la tecnología.

*I spent so much time researching to make a good presentation but because of the technological problems I had to redo the presentation in many forms and platforms. I was sad that beyond my control the video did not work on the day. I don't think this changes my experience of Spanish because I spend so much time thinking on the horrors of technology.*

### 3.2.2. Subtheme: E-Assessments and Time

Comments in this category referred to the student having prepared sufficiently ahead of time (Extracts 18 and 20) for their e-assessment task, the amount of time taken (Extract 26) and planning for next time (Extract 18). Far more often, students commented on the execution of the tasks, i.e., the technical IT e-preparation rather than the quality of their content (Extract 19).

*Extract 18* (I V&R/Report, A): antes que hice la tarea. yo escuche a los documentales españoles sobre el tema. Me ayudo a identificar los puntos clave que había oído. a partir de hora. Habia aprendido que yo necesito afinar mis orejas a español antes de haciendo una tarea. Porque yo puedo oír las palabras importantes.

*Before the task I listened to Spanish documentaries on the topic. It helped me identify the main points I had heard. From then on, I 'd learned that I need to tune my ears to Spanish before doing a task because I can hear the important words.*

*Extract 19* (V&R/Report, D): para prepararme, miré videos en español sobre el tema del comercio justo y leí varios artículos que me ayudarían con las habilidades de lectura y comprensión auditive.

*To prepare myself I watched videos in Spanish on the topic of fair trade and I read various articles that would help me with my reading skills and listening comprehension.*

*Extract 20* (Cult/Essay, C): Además, a escribir un ensayo en un idioma diferente al mío consume más tiempo de lo planeado. En el futuro tendré que planificar al menos el doble de tiempo.

*Also writing an essay in a different language to my own takes a lot more time than I planned. In the future I will plan for at least double the time.*

- RQ 3 How did students orient towards the e-assessment tasks in the reflection?

### 3.3. Theme 3: Student Comments on Personal 'Reflecting'

Generally subsumed into 'reflecting' are personal affective responses and appraisals, in addition to the interpersonal spheres. These sub-themes within the overarching term 'reflecting' are associated with the person and others in and outside the classroom. The personal/interpersonal distinguishes these comments from the other two strands in the analysis: executing the tasks and learning language and culture as course content. They were aware of weaknesses, e.g., Extract 27, on copying too much text in a summary task and when students were assessed on their weaker skills such as listening, they felt insecure (Extract 29). Developing proficiency was also recognized (Extract 26), where a

heritage speaker acknowledges previous limitations in range and Extract 28 where they feel competent and confident after a presentation and are prepared to speak more Spanish.

### 3.3.1. Subtheme: The Personal

Comments in this subtheme contained references to subthemes of awareness of self and capabilities (Extracts 26, 27 and 29) and self-improvement (Extract 28), including the interpersonal such as collaborating with peers (Extracts 29–32), personal fulfilment, plans and meeting personal goals (Extracts 33 and 34).

*Extract 26* (Cult/Essay, J): Esta fue una experiencia que me abrió los ojos, porque esto ha sido mi primera ocasión de redactar parágrafos y textos en español. Con eso me di cuenta que tenía un conocimiento limitado de la lengua. Por haber aprendido el castellano solamente por forma oral y con mi familia.

*This experience was an eye-opener, because this was the first time that I wrote paragraphs and texts in Spanish. Through this I realised that I had a limited knowledge of the language from learning spoken Spanish with my family.*

*Extract 27* (Summary, E): He sido mal en esto antes, porque antes de esa tarea he copiado demasiado del texto original.

*I have been bad at this before because before this task I copied too much of the original text.*

*Extract 28* (Cult/Pres., E): Creo que esta tarea me ha hecho más competente con mi hablar, aún que tenía que mirar mis notas un poco y después la tarea estaba interesada en hablar más en español.

*I believe this task has made me more proficient speaker even though I had to look at my notes a bit and after the task I was interested in speaking more Spanish.*

*Extract 29* (V&R/Report, D): Considero que escuchar y leer son dos de mis habilidades más débiles en el idioma español, por lo que no estaba muy seguro con esta prueba.

*I consider listening and reading to be my weaker skills in Spanish language, so I didn't feel very secure during the test.*

*Extract 33* (Cult./Pres., H): Ahora tengo la habilidad para diseminar argumentos con una confidencia en español que no tenga anterior. Estoy mirando hacia el futuro, donde podía usar esta habilidad en varios escenarios, incluyendo en el salón y en los negocios de las empresas internacionales, por ejemplo, también. Por eso después esta pandemia, yo deseo usar mi conocimiento sobre la industria y la cultura.

*Now I have the ability to make points with a confidence in Spanish that I did not have previously. Looking towards the future where I could use this ability in various settings including rooms of international business for example. Also because of this after the pandemic I wish to use my knowledge of business and culture.*

### 3.3.2. Subtheme: Affective Response

Comments in this subtheme contained references to how the student say they feel. The comments are, by their very nature, typically drawn in relation to affective responses to an e-assessment. Examples include, among others, helping them feel at ease (Extracts 21 and 22), expressing concern about performances on a task (Extract 22), enjoyment or distress (Extracts 23 and 24). The video, reading and writing report was challenging for students because they were required to apply the academic skill of synthesis in Spanish (Extract 22). The culture essay was enjoyed because the topic was current or 'real', as expressed in Extract 24. For the student, it made it interesting to write and gave them a 'good feeling' to be writing so proficiently on an area of interest related to their studies (food sustainability and the environment).

*Extract 21* (Summary, S): Creo que este fue un buen comienzo a las tareas en este curso. En mi opinión, me ayudó poner cómodo el resto del semestre.

*I believe it was a good start to the assessment tasks in the course. In my opinion it helped me feel at ease the rest of the semester.*

*Extract 22* (V&R/Report, E): Esa tarea fue una de las más desafiantes para mí, porque me desafío en mi habilidad de sintetizar información y resumir cosas en mis propias palabras en español.

*This task was one of the more challenging ones for me because I don't have confidence in my ability to synthesize information and summarize into my own words in Spanish.*

*Extract 23* (List/Speak, E): Me pareció muy difícil esta tarea. Hice todo lo posible, pero me resultó especialmente difícil traducir que había escuchado en mis propias palabras a través de hablar, especialmente porque no había reunido toda la información de las grabaciones.

*I found this task very difficult. I did what I could, but it turned out to be especially difficult to translate what I heard into my own words through speaking specially because I had not gathered all the information from the recordings.*

*Extract 24* (Cult/Essay, S): Y por la última fue la evaluación que personalmente fue mi favorita, porque el tema que elegí me parece muy interesante. No sólo esto. El tema elegido es real y me gusta hablar de los eventos que han pasado. Fue un sentimiento muy bueno.

*And for the last assessment that personally was my favourite because the topic I chose seemed very interesting. But more than this the topic chosen was real and I like to discuss events that took place. It was a good feeling.*

3.3.3. Subtheme: The Interpersonal

Comments in this subtheme contained references to student connection with others in their immediate surroundings, such as speaking to personal contacts, Spanish speakers in the community (Extract 25) and in a global context (Extract 34), as well as classroom peers during presentations.

*Extract 25* (Cult./Pres., A): He aprendido a utilizar mis recursos mejor por la próxima vez y uso mis conexiones para darme una comprensión más profunda.

*I have learned for next time, to use my resources better and my [personal] connections to give me a deeper understanding.*

*Extract 30* (Cult./Pres., S): presentar fue muy divertido, en mi opinión, creo que presentar es uno de los métodos más buenos para aprender un idioma. Mi presentación fue bien en general. Creo que era un buen 'jugador' de equipo y en nuestro equipo hicimos una presentación completa y lista también.

*Presenting was fun in my opinion presentations are one of the best ways of learning a language. My presentation went well in general. I believe I was a good team player and also our group made a complete and intelligent presentation.*

*Extract 31* (Cult/Pres., C): La presentación fue la tarea que encontré más agradable. Siempre he tenido muchas ganas de hacer las presentaciones cada semestre de español. Esta tarea no se sintió como una carga. Me sentí competitivo para hacer algo agradable para la clase y mi grupo estaba bien organizado para darme suficiente tiempo para editar los videos.

*The presentation was the most pleasant task. I always really enjoy doing presentations each semester. It does not feel like a weight. I felt competitive to make something enjoyable for the class and my group was very well organised, so I had sufficient time to edit the videos.*

*Extract 32* (Cult./Pres., D): Era la primera vez que trabajaba en grupo con mis compañeros de clase Spanish 5. Nunca los había conocido o hablado con ellos fuera de clase. Pero fue genial conocer a más personas de la clase y trabajar con ellos. No sabía que podía trabajar tan eficientemente como lo hacía con un grupo de personas que nunca había conocido antes. Y fue una experiencia de aprendizaje para mí en la que aprendí que única forma de hacer nuevos amigos es salir de mi 'zona de cómodo muy completo.' Esta tarea me hizo sentir más cómodo con mis compañeros de clase.

*This was the first time that I worked in a group with my Spanish 5 classmates. I had not met them before or spoken outside class. It was awesome to meet more people from class and work with them. I didn't know that one could work as efficiently as I did with a group of people I had never*

*met. It was a learning experience for me, learning that the only way to make new friends completely
out of my comfort zone.*

*Extract 34* (Cult./Essay, J): Yo espero que en el futuro continuaré de usar el español
para mis búsquedas en el ámbito de mis estudios y futuro trabajo.

*I hope in the future I continue to use Spanish in context of searching for studies and future work.*

## 4. Discussion

The study was conceived because there was no research investigating how students
oriented towards L2 language MDCA assessments during the ERT brought on by the
pandemic crisis. In the discussion, as a teacher/researcher, personal reflections are offered
on the findings from this explorative study and how they could be seen to provide evidence
that the students valued e-assessments and attributed learning content, language and
personal skills through the tasks.

In the content analysis of the oral reflection, learners were oriented to all three aspects
of performance addressed in the task rubrics. Their learning of language and culture
through the comparison of skills before and after completing the e-assessment tasks, which
required different skills and the personal side including forward planning. However, this
is not surprising. What is interesting is the delivery of the oral reflection which included
many examples of appraisal and emotion.

The task was designed because the thought of individual oral interviews online, after
a long 12 weeks of ERT, were not something to look forward to. It would not have provided
students with the opportunity to express themselves particularly well, so the five minutes
they might speak within a 10-min interview became the five short reflections. It was also a
convenient way for me as teacher/researcher to reflect on the learner reflections content.

On a task-by-task basis, the reflections show that the students were engaged with the
tasks and the content. In the first summary task they valued being able to search the net in
Spanish for an interesting topic, noting language and structures useful for a summary which
was then assessed. In the second task where learners responded to listening comprehension
questions orally by posting audio, their comments were on the difficulties of understanding
accents which, by level B2, is made more difficult by the speed. Their reflection was that
they needed to practise more online listening. These two initial tasks were formative
tasks in preparation for the integrated listening task later in the semester which combined
authentic news videos and online news text. Some students commented that the tasks set
them up for the semester with an idea of the expected standard and what to work towards,
which meant working with authentic texts online.

The reflection on academic skills development and appreciation of the cultural compo-
nents of the assessment came through very strongly, particularly underlined by appraisals
and emotive comments. It was pleasing to see how the comments showed a strong com-
mitment from the learners to the videoed PowerPoint visual presentation as a measure of
academic skills. The comments demonstrated that they found it not only useful in terms of
the ability to produce relevant, well-organized, and meaningful content, but also the extent
to which it showed that the student had carried out e-research in Spanish, which had been
synthesised into their recorded voiceover in a way that other students could understand
the topic being presented accompanied by visuals. This perspective concurs with that
of Joughin (1998, p. 368), who describes academic oral assessment as being where 'the
object of assessment is not the oral ability of the student but rather the student's cognitive
knowledge, understanding, thinking processes, and capacity to communicate in relation
to these'.

The presentation was to have minimal writing, but visuals supported comprehension.
Students commented on their selection of content that was engaging or relevant to the
other student listeners as the audience. The use of visual support materials to support
understanding has been found in analyses of academic presentations in other studies, e.g.,
Jacoby and McNamara (1999). Audience awareness was invoked as being valued in the
student reflections about being creative and keeping the audience engaged.

Additionally, the comments on the culture essay assessments showed that in addition to the presentation skills, in the Spanish class at B2 level, students were aware of their academic skill development. Similar comments were made about carrying out e-research in Spanish, but at a deeper level was the comment that reading the texts provided a model for Spanish student essay writing. The reflections on conducting e-research, writing up and presenting on the culture via the presentation and essays can be inferred as awareness by the students of socialisation into academic culture (cf. Duff 2007, 2010). Language use surprisingly proved not to be the focus of attention, considering it is a Spanish course; however, it may indicate that the level B2 students are working on proficiency and not discrete skills, which emerges from the reflections.

The results from the content analysis of the student reflections discussed above are examples providing evidence that language testing and assessment during the pandemic and within an ERT context was an exceptionally differently situated activity, located in the pandemic context, characterized by unplanned and quickly adapted assessment practices and policies. The students' social and pedagogical values and beliefs drawn from the data analysis show that as a local community, they uphold learning from their peers, are learning from research online, and have plans to improve and use their skills in the future.

The reflections provided evidence that the qualities students attribute to e-assessments showed they valued by them. Despite the surprise of Emergency Remote Teaching (ERT) and e-assessments that were designed for authenticity and test security on the run, the learner reflections provide high-order reflections that make explicit to language educators and course designers that learners are skilled and willing to discuss their assessment performance explicitly and candidly in light of their perceived assessment requirements. As a teacher/researcher observing the students, they appeared to be working towards the same learning outcomes as were intended, but what was surprising was the candid expression that accompanied the oral reflections. Two students commented on the oral reflection activity as part of their five tasks. (Although it was not a listed option, it was interesting to read.) They commented that the speaking reflection mode made the reflection seem more authentic and less 'prepared' and that reflection as a task helps learners evaluate all that has been achieved in a semester.

The wider implication that this study brings to the field is the implementation of alternative assessment practices in L2 instruction, which could also take place within F2F or online, non-ERT, teaching environments. This exploration in the ERT context provides an insight into student-centred learning occurring in various locations outside educational institutions globally. Learners worked without the support of mentors or classmates, so analysing the content of their reflections on their achievements and future plans to improve was a unique way of gaining insight into L2 adult language learning needs and processes in educational emergency situations.

## 5. Conclusions

The aim of this paper was to report on student orientation to e-assessments through an oral reflection task in Spanish. The learners' own reflections have offered perspectives on their study practices, perceptions, and knowledge about learning through re-designed MDCA assessment tasks in an emergency remote teaching context. At the heart of the reflections lie the learners' understanding of learning and assessment goals and how they are achieved. These valuable lessons from learners taking MDCA assessments for the first time in the context of ERT may be helpful for future remote teaching events in the post-COVID-19 era.

**Funding:** This research received no external funding.

**Institutional Review Board Statement:** The study was conducted in accordance with the Declaration of Helsinki, and approved by the Human Research Ethics Committee of RMIT University (protocol code 2020-23688-13090) approved 3/12/20-2023.

**Informed Consent Statement:** Informed consent was obtained from all subjects involved in the study.

**Data Availability Statement:** Data are stored at RMIT University following ethics protocol.

**Conflicts of Interest:** The authors declare no conflict of interest.

## Appendix A. The Reflection Task Instructions

**Personal Language Learning Reflection Guide: What Shall I Say?/What Is Marked?**
1. Focus Of Oral Reflection

- **Learning Experiences**—You must identify and define **five significant concrete learning experiences** in relation to the Spanish 5 course content/assessments and use vocabulary and structures acquired to reflect in Spanish.
- **Personalization**—Personalize your learning experiences and relate these to yourself. Consider: How personal is the statement? Are you telling your story or an objective account? A good statement will contain mainly 'I' statements.

2. Ability to Reflect

- **Describe the starting point:** Describe, recall, define and review your experiences. You cannot reflect on how learning experiences affected you, if you have nothing with which to compare.
- **(Describe the experiences and the effect(s) on the initial position:** Break down experiences by examining, questioning, comparing, and criticizing. Talk about the experiences but not by making a series of vague general unconnected statements. Consider: Have you described and analyzed the experience? Have you painted a picture? What have you learnt from the experiences? Did your initial thinking change or not as a result of these experiences?
- **Summarize the present position:** On drawing a conclusion, justify the statement by saying why you have reached that conclusion. Consider: what have you decided as a result of your analysis of the learning experiences and why?
- **Outline how the experience has influenced your outlook:** Integrate what has been learned into forward planning and future goals. Make connections, i.e., to combine what has been learned from the different experiences into a plan

## Appendix B. The List of Tasks and Reflection Criteria

quiz 1 e-reading > grammar & vocab selection > summary writing
quiz 2 audio listening > voice memo recorded speaking
Integrated listening task e-reading & video listening > report writing
presentation of online culture research > written & visuals with recorded oral on PPT
quiz 3 culture reading, audio comprehension, grammar, writing > drafted using online editing tools
culture essay e-research > essay writing

| *FOCUS OF ORAL REFLECTION 10%* | *Total* |
| --- | --- |
| *Task completion* 5 min of oral in Spanish; completing and submitting reflections on learning; the process of recording and submitting 5 × 1 min sound files; reflecting using guide on any 5 tasks listed, choosing best one. *TEACHER MARKS your best one, e.g., most personalisation* | /4 |
| *Ability to include language and structures studied* | /4 |
| *Ability to Reflect:* *(a) Describes the starting point in learning* *(b) Describes the experiences and the effect(s) learning and assessment* *(c) Summarise the present following feedback and marking* *(d) Outline your outlook for future plans following the assessment* *Feedback comment box* | /3 /3 /3 /3 |

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
