# Peer review of "Oral Reflection Tasks: Advanced Spanish L2 Learner Insights on Emergency Remote Teaching Assessment Practices in a Higher Education Context"

_languages, doi:10.3390/languages7010026_

Round 1
Reviewer 1 Report
Review report
Paper title:
Oral reflection tasks: Advanced Spanish L2 learner insights on Emergency Remote Teaching assessment practices in higher education context
The study intends to explore advanced Spanish L2 learners’ orientation towards e-assessment reflective practices in an Emergency Remote Teaching (ERT) context, which was imposed because of the covid-19 pandemic restrictions. In this sense, the study can be considered original, since there has been scarce research addressing this new situation of teaching and assessment of foreign languages in higher education. Another significant contribution of this paper relates to the examples and information provided in terms of the learners’ reflection experience, which could function as practical guidance to L2 instructors that wish to promote the development of reflection skills in their students, not only at the tertiary level of education but also at other contexts of instruction. In what follows, there are some suggestions for further improvement, addressing each section in separate.
- Introduction
The rationale of the study has been adequately contextualized with respect to previous research, especially in the fields of ‘emergency remote teaching’ (ERT) and reflection practices applied in educational contexts. However, the reflective skills that are expected to be promoted within an L2 learning environment could also be related to the promotion of 21st century skills, which are mentioned at the end of this section (lines 180-181), because this connection could further justify the importance of this research. Reference to 21st skills could potentially be made earlier in this section, provided it is supported by relevant in-text citations, which are currently missing.
Regarding the use of sources in this section, it seems that in line 80 the source mentioned should be further elaborated, because it is not very clear how the aim of the present study relates to the research by Vogt et al (2020). This addition could also strengthen the argument made in this paragraph.
- Introduction and Results
In relation to the research questions (lines 94-96), the second research question could be further specified (e.g. What was learned… in terms of …?). Moreover, the third question seems a bit vague and could be further specified as well.
A more general comment regarding the research questions relates to their answer. In particular, in the ‘Results’ section there is not a clear correspondence between the way the findings are presented and how they answer the research questions posed in the ‘Introduction’. This could be improved if the ‘Results’ section included a different grouping of the transcribed extracts, so that the answers to the research questions were more direct and clear. Alternatively, these answers could be added as comments in the ‘Discussion’ section.
- Introduction and Methodology
The option of transferring the information presented in sub-sections 1.2. and 1.3 to Section 2 (Materials and Methods) should be considered, because the information about the research context (setting, established teaching and assessment practices), as well as the description of the course assessment tasks used as part of the research implementation, typically constitute elements of the research method.
Line 174: Was the task marking rubric (appendix 2) based on specific criteria for reflective practice set in previous studies? The addition of relevant research would strengthen the validity of the marking rubric.
2.Materials and Methods
2.3. Data collection
Line 218: It is not very clear how some submissions were considered as ‘partly irrelevant’.
Table 2 and lines 221-229:
The comments made under Table 2 should follow the order in which the information is presented on the table, so that clarity can be increased. In particular, the multimodal task types could be presented on the table in the same order as the comments made under the table. Alternatively, the task types could be presented on the table in order of frequency (e.g. higher to lower). Moreover, Table 2 should follow (and not precede) all the relevant comments made in the text.
In lines 221-226, the information could be rephrased, in order for more emphasis to be placed on the types of tasks that gathered a higher percentage. The division of this part into two separate sentences could be tried.
Lines 226-227: unclear information and phrasing. To which multimodal task (Table 2) do the ‘summative language and culture final’ correspond to?
Lines 227-229: this comment should be rephrased to sound more formal. Another option would be for this observation to be mentioned in the ‘Discussion’ section only.
2.4. Analysis of the transcribed oral reflection tasks
This part of the methodology should include some references to similar research, in which analysis of qualitative data had been conducted. Relevant sources should also be used to support the concepts mentioned in line 243.
3.Results
Line 266: the “three overarching categories” should be clearly named, maybe within parentheses.
Lines 302-303: the information should be rephrased for more clarity.
General comment for the ‘Results’ section:
The rationale behind the division of the three overarching categories into sub-themes is not very clear. The merging of some sub-categories that present common characteristics could be considered (e.g. subtheme 3.1.3.2 could be merged with 3.1.3), in order for clarity and comprehensibility to be increased. Moreover, there should be more analysis on some transcribed extracts, whose length could probably be shortened. For example, in subtheme 3.1.1. (Affective response) there could be some comments on how exactly the students felt about specific tasks, that is, which task was “more challenging”, “difficult”, “interesting”, “real” and how these perceptions related to their feelings of “confidence”, “enjoyment” or “distress”. In the same way, in 3.2 (Theme 2), there should be more specific comments on the challenges the students faced and the skills they had the chance to practise. More specific comments are also needed in subthemes 3.3.1.1. and 3.3.2.
An alternative way of organizing the ‘Results’ section would be the following: the content of the students’ reflection could be analyzed in two basic dimensions “personal” and ‘interpersonal” in relation to the three overarching categories.
Overall, it should be noted that this section should directly answer the research questions mentioned in the ‘Introduction’ (lines 94-96).
- Discussion – Conclusion
There might be some additional analysis regarding the wider implications of this study in the field of alternative assessment practices in L2 instruction, which could probably take place within both live and online teaching environments. Implications could also be considered in relation to student-centeredness and the exploration of adult learners’ needs, as a means of facilitating the L2 learning process in emergency situations.
Text formality:
Some revisions could probably be made in an effort to reduce the use of the personal pronoun ‘I”, especially in the ‘Discussion’ section.
Reference to source material:
Lines 58-59: “Brockbank and McGill, 1998”: not all three editors are mentioned
Line 62: ‘Huang, 2007’: source not included in the ‘References’
Line 84: page mentioned before the authors
Line 148: “Pal and Koris, 2021”: In the ‘References’ the authors are mentioned in a different order.
Line 202: “Koshy and Drane, 2021”: source not included in the ‘References’
Lines 733-734: This source was not found as an in-text citation.
The guidelines regarding the way sources of more than two authors are cited within text should be revised, because there has been some inconsistency (e.g. Swain et al. (line 62) vs Vogt, K., Tsagari, D. & Csepes, I. 2020 (line 80)).
Reviewer 2 Report
Some categories must be more explained.
Conclusions must be improved.
References about SFL (ELE) must be included, and also Spanish authors.

Reviewer 3 Report
Literature Review: The literature review is very underdeveloped. The author provides some information about the use of oral reflection tasks and some information about assessment but provides no background or context on the findings of previous studies that have used oral reflection tasks in other contexts; that is, contexts other than emergency remote teaching. Therefore, while the author claims that the study addresses a new question and contributes new information, the author fails to situate the study in the broader context of previous findings in related studies.
Methodology and Analysis. I have included extensive notes and suggestions in the manuscript itself. The main concerns I will note here are the following:
- The author claims in the discussion that the study provides "evidence that the qualities students attribute to e-assessment showed they valued by (sic) them." While this is a qualitative study, some additional percentages would have certainly helped justify this claim. The extracts showed both 'positive' and 'negative' opinions of various aspects of the tasks on which the participants were reflecting. It would be useful to group together all of the 'positive' comments (by theme, if necessary) and all of the 'negative' comments to then get an overall sense of how much these e-assessment qualities were valued by the participants.
- Again, the lack of information about the findings of other studies using this methodology in other, non-ERT contexts, make it difficult to understand how the findings of this study should be understood.
- I believe asking the participants to reflect on tasks that they completed weeks before, and to reflect on five different tasks based on the long-term memory of carrying out each task, may pose a big limitation in the methodology of this study. The reflections do not necessarily reflect immediate reactions to the e-assessments. They may also carry a variety of other thoughts and feelings about various aspects of the course in general, or some final sense of the entire course in general, since they were reflecting on these assessments after the course had ended.
- A more thorough description of each of the tasks on which the participants reflected is needed in the "Materials" section. The entire Methodology section lacks flow and cohesion with respect to materials and procedure.
- The very first instruction in Appendix A were not easy to understand. I imagine the participants may have had difficulties, too.
Referencing: The author must become more familiar with practice of adequate use of references in the text. In more than one occasion, which I have highlighted in the text, it is either not clear why the author references a study at all or why the author references a study in that particular place in the sentence.
Writing: style, grammar: Unfortunately, the entire manuscript is filled with syntax errors and stylistic problems and inconsistencies (e.g., the use of "i.e.", the use of punctuation, sentences lacking a main verb). Additionally, the entire introduction/literature review lacks flow and cohesion; it reads like disconnected bits of information pieced together rather than an organized set of properly-developed topics.
Conclusion: This study does have potential but it will require a lot of work to get it to publishable quality. The syntax, grammar and writing style must be revised extensively. The literature review must be developed a lot more. The limitations of the methodology must be addressed. The presentation of the results must be improved. The study must be adequately situation in a broader context. The claims with respect to "evidence" must be put forward cautiously. Please see the manuscript for all my comments and suggestions.

Round 2
Reviewer 1 Report
The revised draft provides evidence that the author has taken into consideration most of the suggested comments made by the reviewers and made the respective changes. A final suggestion relates to the change of certain lexical choices, especially in the 'Discussion' section, so that the formal style of the text can be enhanced. For example, lines 726-728, the replacement of the words "bedrooms, gardens,studies loungerooms", "buddies" should be considered.
Author Response
This exploration in the ERT context provides an insight into student centred learning occurring in various locations outside educational institutions bedrooms, studies, gardens and loungerooms globally. Learners worked without the support of mentors buddies or classmates, so analysing
Thank you for reminding me to raise the tone of the expression here!
It is deleted and replaced in the round 2 revised manuscript attached highlights for you in red.
cheers
Reviewer 2 Report
The author has made a great effort to improve the paper.
Author Response
Thank you - no changes requested.
cheers
Reviewer 3 Report
There is good evidence that the author made substantial revisions to the manuscript. There are some lingering grammar and stylistic issues. Not all of the comments/requests for revision of Reviewer 3 were addressed.

Author Response
Apologies for the oversight - it was unintentional.
all comments have been responded to in pdf and then have b een copied in green (for reviewer 3 ) into the final manuscript attached.
Thank you so much.
